# Spirulina-enriched Substrate to Rear Black Soldier Fly (*Hermetia illucens*) Prepupae as Alternative Aquafeed Ingredient for Rainbow Trout (*Oncorhynchus mykiss*) Diets: Possible Effects on Zootechnical Performances, Gut and Liver Health Status, and Fillet Quality

**DOI:** 10.3390/ani13010173

**Published:** 2023-01-02

**Authors:** Stefano Ratti, Matteo Zarantoniello, Giulia Chemello, Miriam Giammarino, Francesco Alessandro Palermo, Paolo Cocci, Gilberto Mosconi, Maria Vittoria Tignani, Giulia Pascon, Gloriana Cardinaletti, Deborah Pacetti, Ancuta Nartea, Giuliana Parisi, Paola Riolo, Alessia Belloni, Ike Olivotto

**Affiliations:** 1Department of Life and Environmental Sciences, Polytechnic University of Marche, 60131 Ancona, Italy; 2School of Biosciences and Veterinary Medicine, University of Camerino, 62032 Camerino, Italy; 3Department of Agriculture, Food, Environment and Forestry, University of Florence, 50144 Firenze, Italy; 4Department of Agricultural, Food, Environmental and Animal Science, University of Udine, 33100 Udine, Italy; 5Department of Agricultural, Food and Environmental Sciences, Polytechnic University of Marche, 60131 Ancona, Italy

**Keywords:** sustainable aquaculture, fish nutrition, insects, distal intestine, salmonids

## Abstract

**Simple Summary:**

To promote sustainability in aquaculture, the black soldier fly deserves special attention as an alternative ingredient for aquafeed formulation. The present study proposed the inclusion of spirulina in the growth substrate of black soldier fly prepupae to enrich their final biomass in terms of polyunsaturated fatty acids and antioxidant molecules. The obtained prepupae meal was used as a replacer of unsustainable marine-derived ingredients in diets intended for rainbow trout during a 6-week feeding trial. The results showed that fish zootechnical performances, gut and liver health status, and marketable characteristics were not negatively affected by the experimental diets.

**Abstract:**

In the present study, an organic substrate (coffee silverskin) enriched with spirulina (*Arthrospira platensis*; 15% *w*/*w*), as a source of lipids and bioactive molecules, was used to rear the black soldier fly (*Hermetia illucens*) prepupae. Three grossly isonitrogenous, isoproteic, isolipidic and isoenergetic experimental diets for rainbow trout (*Oncorhynchus mykiss*) juveniles were then produced: a control diet (HM0) mostly including fish meal and fish oil, and two other test diets named HM3 and HM20, in which 3 or 20% of the marine ingredients were substituted with full fat black soldier fly prepupae meal (HM), respectively. Experimental diets were provided for 6 weeks, and at the end of the trial the physiological responses and marketable traits of the fish were investigated using a multidisciplinary approach. Generally, all test diets were well accepted, and fish growth, gut and liver health status, and marketable characteristics were not impaired by the experimental diets. However, an increased immuno-related gene expression along with a slight reduction of fillet redness and yellowness was evident in fish from the HM20 group.

## 1. Introduction

Over the last decades, the aquaculture sector has been interested in the reduction of unsustainable fish meal (FM) and fish oil (FO) in aquafeed formulations to promote alternative and more sustainable ingredients [1,2,3]. Subsequently, a number of new ingredients have been searched and tested over the last years, including vegetable ingredients [4], processed animal proteins (PAPs) [5,6] and dried microbial biomass from microalgae [7,8,9,10,11].

More recently, insect meal has gained attention in aquaculture nutrition [2,12,13,14,15]. In this context, the black soldier fly (*Hermetia illucens*) has been widely considered [16,17,18,19,20,21] due to (i) a proper protein content (approximately 32%) [22] and an essential amino acid (EAA) profile close to FM [23,24,25]; (ii) a high feed-conversion efficiency, substrate consumption and waste reduction index of the larvae [26,27]; (iii) the ability to grow on different organic by-products [28,29]; and the low environmental impact of rearing culture (i.e., land use, water consumption and CO_2_ production) [30,31]. These properties allow the application of the circular economy concept to the aquaculture sector, using for example land-produced organic by-product for insect rearing. Furthermore, it is well established that black soldier fly larvae/prepupae naturally contain different molecules (i.e., chitin, antimicrobial peptides, and lauric acid) that can play an immunostimulant and/or anti-inflammatory role in fish [15,32,33,34].

However, regarding the lipid content, the fatty acids (FA) profile of black soldier fly prepupae meal (HM) is characterized by a low level of long-chain polyunsaturated fatty acids (PUFA) and a high quantity of saturated ones (SFA) [33,35,36,37]. Nevertheless, the FA profile of the larvae can be partially modulated by their diet [38,39], potentially overcoming the drawback related to their unbalanced SFA/PUFA ratio [40,41,42,43]. Hence, the inclusion of the dried biomass from a marine protist rich in long-chain PUFA (*Schizochytrium* sp., 10% *w*/*w*) in the growth substrate of black soldier fly larvae has been shown to successfully improve the PUFAs content of the insects’ final biomass [39].

On this regard, the dried biomass of spirulina (*Arthrospira platensis)* can represent an equally valid supplement for the insect growth substrate because of its nutritional values [44,45]. In fact, spirulina, the most commonly cultured cyanobacteria produced at commercial scale [46], shows a high content of PUFAs for fish requirement [47,48]. Furthermore, spirulina contains several compounds such as vitamins, minerals, tocopherols, and carotenoids which can have a positive antioxidant effect on fish [49,50,51].

The low inclusion (1 to 15%) of spirulina in fish diets has already been tested in different studies [52,53,54] showing that it is able to improve fish growth, survival rate, feed intake, and fillet color and firmness. However, higher dietary inclusion levels led to controversial results regarding fish-growth performances, mostly related to a reduced feed acceptance [55,56,57,58]. In addition, the 9% dietary inclusion of spirulina in diets totally deprived of FM was not able to fully counteract the adverse effects of certain plant protein-rich ingredients on the gut health status of rainbow trout (*Oncorhynchus mykiss*) [59].

These results, in addition to its high production costs [60], suggest that spirulina, instead of an aquafeed ingredient or supplement in aquafeed formulation, could be used to enrich the growth substrate of black soldier fly larvae, giving the possibility to exploit its beneficial properties avoiding a direct inclusion in fish diets. This aspect is also crucial to avoid the regularly used defatting processes applied to insect meal, which is one of the main reasons of the high final costs of this new ingredient.

For these reasons, this study aimed to enrich the insects’ rearing substrate (in terms of PUFAs and antioxidants molecules contents) with a 15% *w*/*w* inclusion of spirulina and then use the obtained full-fat prepupae meal to replace 0, 3 or 20% of FM and FO in experimental diets (named HM0, HM3, and HM20, respectively) intended for rainbow trout (which represents 53% of the total freshwater European aquaculture production) [61]. The two selected inclusion levels were chosen according to the range used over the last year in studies on rainbow trout [62]; this was because 20% represents an ecological inclusion level (a global 20% FM substitution in aquafeeds has a proper ecological impact), while the 3% substitution level should be considered as a feed supplement inclusion with possible ameliorative effects on fish health status. A multidisciplinary approach was then applied to evaluate fish zootechnical performances, gut and liver health status, and quality-marketable characteristics.

## 2. Materials and Methods

### 2.1. Ethics

Fish manipulation was performed following the procedures established by the European Communities Council Directive (86/609/EEC and 2010/63/EU) for animal health and approved by the Camerino’s University Animal Ethics Committee (Approval Number 6/2021). Optimal rearing conditions were ensured, and animal suffering was minimized using the MS222 anesthetic (Merck KGaA, Darmstadt, Germany).

### 2.2. Insects’ Rearing and Production of Fish Diets

Full-fat black soldier fly prepupae were obtained after rearing phase performed at the D3A Department of the Marche Polytechnic University (Ancona, Italy). Briefly, black soldier fly larvae were reared for 21 days in a climatic chamber (27 ± 1 °C; 65 ± 5% relative humidity) in continuous darkness following the methods adopted by Spranghers et al. [63]. Larval rearing was performed on a growth substrate based on 85% of coffee silverskin and 15% (*w*/*w*) of spirulina dry biomass, adding distilled water to obtain ~70% of final moisture [23]. Rearing density was 0.3/cm^2^ [64] and the feeding rate per larva was 100 mg/day [65]. The growth substrate was replaced once a week until the prepupal stage was reached, identified by the color change of the integument from white to black [66]. At the end of the rearing phase, prepupae were freeze dried and ground to obtain an insect meal (HM) for the subsequent fish diets formulation. Three complete diets were prepared to cover all the nutrient requirements for rainbow trout [67]: (i) a control diet (HM0) based on conventional marine (FM and FO) and vegetable ingredients, and (ii) two test diets obtained from the HM0 formulation by replacing 3 (HM3) or 20 (HM20) % of FM and FO with HM (1.8 and 12% levels of dietary inclusion, respectively). All ingredients were ground (0.5 mm) with a Retsch Centrifugal Grinding Mill ZM 1000 (Retsch GmbH, Haan, Germany) and well-mixed with FO to form a homogeneous blend (Kenwood kMix KMX53 stand Mixer). The resulting meshes were added with water (~350 g/kg) and the doughs, thus obtained, were cold extruded into 3 mm pellets using a meat mincer provided with a knife at the die. The wet pellets were then dried in a ventilated oven at 37 °C for 48 h. The diets obtained were stored in sealed bags under vacuum and kept at 4 °C until use. The HM and the experimental diets were analyzed in duplicate for dry matter (DM), total nitrogen (CP) by using the Kjeldahl method, ether extract (EE), and ash content following the AOAC [68]. The gross energy content (GE) was determined using an adiabatic calorimetric bomb (IKA C7000, Werke GmbH & Co., Staufen, Germany). For the N determination, a nitrogen-to-protein factor of 4.67 was used, as suggested by Janssen et al. [69], only for the insect ingredient used for diet formulation. In addition, the chitin content of HM and experimental diets was determined according to the method described by Hahn et al. [70]. The dietary ingredients and proximate composition of both HM and experimental diets are reported in Table 1.

### 2.3. Total Lipids, Fatty Acids, Carotenoids and Tocopherol Determination in Fish Diet

Total lipids were isolated as described by Folch et al. [71]. Minced freeze-dried diets (2 g) were dissolved in 2:1 chloroform to methanol (*v*/*v*, 40 mL), with added tridecanoic and nonadecanoic acid methyl esters as internal standards (500 μL of a 10 mg/mL solution in *n*-hexane), agitated for 5 min, and centrifuged (3000 rpm, 10 min, 4 °C). The organic phase was washed with distilled water (5 mL), filtered through Whatman filter paper (Grade 4, 90 mm, Merck KGaA, Darmstadt, Germany) over anhydrous sodium sulphate (3 g) and evaporated with a rotary evaporator (30 °C) to collect the fat.

Fatty acids methyl esters (FAME) were obtained from total lipids through transmethylation by a BF_3_-MeOH reagent [72]. Briefly, 20 mg of fat were added to *n*-hexane (0.5 mL) and a BF_3_-MeOH solution (0.5 mL) and vortexed. After 15 min at 100 °C, the reaction was interrupted with distilled water (0.5 mL), and the mixture was centrifuged (4500 rpm, 3 min). Capillary gas chromatography was used to analyze the organic phase as reported by Balzano et al. [73].

Carotenoids were extracted from freeze-dried samples of diets and analyzed by liquid chromatography as reported by Nartea et al. [74]. Quantification of carotenoid was performed by external calibration and, in all cases, a correlation coefficient of 0.999 was obtained. For zeaxanthin, the instrumental limit of detection (LOD) and quantification (LOQ) were 5 and 16 ng/mL, respectively, while for β-carotene it was 5 and 18 ng/mL, respectively.

For tocopherol determination, 100 mg of freeze-dried samples were added to 5 mL of hexane, vortexed for 5 minutes, and centrifuged (3500 rpm, 2 min). The organic phase was collected. The extraction was repeated a second time and the organic fractions were pooled together, filtered (0.45 μm, Sartorius Regenerated Cellulose Membrane), dried, resuspended in 0.5 mL hexane, and injected in a Waters Ultra Pressure Liquid Chromatographic Acquity system (UPLC Acquity H-Class, Waters Corporation, Milford, CT, USA) equipped with a Fluorimetric Detector (FLD) and an Ascentis Express Hilic column (15 cm × 2.1 mm, 2.7 mm) [74,75]. An isocratic elution of *n*-hexane, isopropanol, and acetic acid (95.5, 0.4, 0.1%, respectively) at 0.3 mL/min was performed at a temperature of 30 °C (column heater and sample loading). Tocopherols were detected by comparison of retention time with pure standards and quantified with external calibration. The calibration curves of α-, γ- and δ-tocopherol ranged from 3 to 100 mg/mL with correlation coefficients higher than 0.986. The instrumental LOD and LOQ were 4 and 14 ng/mL for α-tocopherol, 2 and 7 ng/mL for γ-tocopherol, and 2 and 7 ng/mL for δ-tocopherol. Vitamin E content was calculated considering only α-tocopherol as proposed by EFSA NDA Panel.

### 2.4. Fish Rearing Conditions and Sampling

Six hundred rainbow trout juveniles (15.3 ± 2.2 g) were provided by Itticoltura Valpotenza snc (Fiuminata, MC, Italy) and were maintained for a two-week acclimatization period at “Unità di Ricerca e Didattica di San Benedetto del Tronto (AP, Italy), URDIS-University of Camerino” in 4 m^3^ tanks with a mechanical and a biological filtration system (Scubla aquaculture snc, Remanzacco, UD, Italy). The temperature was maintained at 16.2 ± 0.2 °C by chillers (Scubla aquaculture snc, Remanzacco, UD, Italy) and pH values were kept constant at 7.88 ± 0.30.

At the beginning of the trial, fish were lightly anesthetized (150 mg/L of MS222; Merck KGaA, Darmstadt, Germany) and individually measured. Five hundred forty fish were distributed into nine fiberglass square tanks (4 m^3^ each; three tanks per dietary group, 60 fish per tank) that were maintained at the same chemical-physical conditions and were equipped with the same filtration system described for the acclimatization phase. For 6 weeks, fish were fed the experimental diets provided in 1 daily meal in the morning until apparent satiety. All the feed provided was completely ingested by the fish within 30 min after feeding.

After the 6 weeks of trial, fish were sacrificed with a lethal dose of MS222 (1 g/L) and individually measured. Samples of liver, pyloric caeca, distal intestine, and whole fish (residual to the previous samplings) were collected and properly stored for further analyses.

### 2.5. Growth Performance, Condition Factor, Biometric and Marketable Characteristics

Final survival rate was calculated by removing the number of dead fish from the initial number of specimens. Biometric measurements (final body weight -FBW- and standard length), growth performance [relative growth rate (RGR), specific growth rate (SGR), Fulton’s condition factor (K)], and feed utilization (FCR) were calculated per fish (*n* = 60), as follows:RGR (%) = [FBW − IBW (initial body weight, g)] × 100/ IBW(1)
SGR (%/day) = [(ln FBW − ln IBW)/number of feeding days] × 100(2)
K = [FBW (g)/standard length^3^ (cm)] × 100(3)
FCR = Feed intake (g)/biomass gain (g)(4)

Marketable characteristic [fillet yield (FY), hepatosomatic index (HSI), carcass weight (CW) and viscerosomatic index (VSI)] were calculated per fish (*n* = 15), as follows:FY (%) = [(fillet with skin weight (g)/body weight (g)] × 100(5)
HSI (%) = [(liver weight (g)/body weight (g)] × 100(6)
CW (%) = [(carcass weight (g)/body weight (g)] × 100(7)
VSI (%) = [(viscera weight (g)/body weight (g)] × 100(8)

### 2.6. Histological Analysis

Distal intestine and liver samples from 5 fish per tank (*n* = 15) were fixed for 24 h at 4 °C in Bouin’s solution and then processed according to Cardinaletti et al. [13] and Randazzo et al. [76]. Briefly, samples were washed with ethanol (70%), dehydrated in graded ethanol solutions, washed with xylene, and finally embedded in paraffin (Bio-Optica, Milano, Italy). A Leica RM2125 RTS microtome (Nussloch, Germany) was used to cut sections of 5 µm thickness that were stained with Mayer hematoxylin and eosin Y (Merck KGaA). Histological evaluations were conducted by three operators through independent blind evaluations using a Zeiss Axio Imager.A2 (Zeiss, Oberkochen, Germany) microscope equipped with a combined color digital camera (Axiocam 105, Zeiss). Image analyses were performed using the ZEN 2.3 software (Zeiss).

Distal intestine. For each sample, 6 transversal sections were collected at 200 μm intervals and analyzed to measure mucosal fold height, submucosa width, goblet cells and supranuclear vacuoles according to Randazzo et al. [77].

Regarding the evaluation of the goblet cell’s relative abundance, cells were counted on 300 μm^2^ of absorptive surface on each section. For supranuclear vacuole analysis, an arbitrary index score was assigned as follows: + = scattered, ++ = abundant) [78].

Liver. Three sections from each sample collected at 200 μm intervals were analyzed to evaluate hepatocyte morphology and the structure of hepatic parenchyma.

### 2.7. Vibrational Spectroscopic Analyses

Liver sample preparation and spectroscopic measurements using Fourier Transform InfraRed Imaging (FTIRI) analysis were carried out according to the literature [79,80,81,82]. After sampling, liver samples from three fish per tank (*n* = 9) were immediately stored at −80 °C. For each sample, three 10 μm sections were collected at 200 μm intervals using a cryotome. Then, sections were deposited on CaF_2_ optical windows (13 mm diameter, 1 mm thick) and then air-dried for 30 min before the acquisition. A Bruker INVENIO-R interferometer coupled with a Hyperion 3000 Vis-IR microscope and equipped with a Focal Plane Array (FPA) detector operating at liquid nitrogen temperature (Bruker Optics, Ettlingen, Germany) was used for FTIRI measurements. For each liver section, a 15X condenser/objective was used to detect specific areas in which the IR maps (164 × 164 μm size; 4096 pixel/spectra with a spatial resolution 2.56 × 2.56 μm) were acquired in the middle infrared range (4000–800 cm^−1^) and in transmission mode, performing 256 scans with a spectral resolution of 4 cm^−1^. A background spectrum was acquired on a clean portion of the CaF_2_ optical window before each acquisition. Raw IR maps were submitted to spectral pre-processing to correct the water vapor and carbon dioxide contributions and to remove thickness variation artifacts on the full frequency range (respectively through Atmospheric Compensation and Vector Normalization routines; OPUS 7.5 software package).

To obtain the false color images, the pre-processed IR maps were integrated in the following spectral ranges: 3000–2800 cm^−1^ (representative of lipids, LIPIDS), and 1710–1480 cm^−1^ (representative of proteins, PROTEINS). Then, the band area ratios LIP/TBM (area of the 3050–2800 cm^−1^ region divided for the sum of the areas of the 3050–2800 cm^−1^ and 1760–950 cm^−1^ regions, named TBM), FA/TBM (area of the 1780–1700 cm^−1^ region, representative of fatty acids, divided for TBM), and PRT/TBM (area of the 1700–1480 cm^−1^ region divided for TBM) were calculated and statistically analyzed.

### 2.8. RNA Extraction and cDNA Synthesis

Total RNA was extracted from the liver, distal intestine and pyloric caeca from 5 fish from each tank (*n* = 15) using RNAzol RT reagent (Merck KGaA), eluted in 20 µL of RNase-free water (Qiagen), and then stored at −80 °C until use [83]. A NanoPhotometer P-Class (Implen, München, Germany) was used to determine final the RNA concentration of each sample. RNA integrity was verified by GelRed^TM^ staining of 28S and 18S ribosomal RNA bands on 1% agarose gel. A cDNA synthesis was performed using the High-Capacity cDNA Reverse Transcription Kit (Bio-Rad, Milano, Italy) from 1 μg of RNA.

### 2.9. Real-Time PCR

PCR analyses were performed using an iQ5 iCycler thermal cycler (Bio-Rad, Milano, Italy) following the method reported in Maradonna et al. [84]. For all the reactions, the thermal profile started with 3 min at 95 °C, and then was characterized by 45 cycles of 20 s at 95 °C, 20 s at 60 °C, and 20 s at 72 °C. At the end of each cycle, the fluorescent signal was detected. The melting curve analysis confirmed, for each reaction, the presence of only one PCR amplification product, except for negative controls that did not reveal amplification products and primer-dimer formation. Amplification products were sequenced, and homology was verified.

Quantification of relative mRNA abundance was performed in (i) liver samples for genes involved in stress response [glucocorticoid receptor (*gr*), heat shock protein 70 (*hsp70*)] and fish growth [insulin-like growth factor-1 (*igf1*), growth hormone receptor-1 (*ghr1*), myostatin1a (*mstn1a*)] was performed on liver samples; (ii) pyloric caeca samples for genes involved in PUFA biosynthesis [fatty acid elongase 2 (*elovl2*), fatty acid desaturases 2 (*fads2*)] (according to Bruni et al. [85]); (iii) in distal intestine samples for genes involved in the inflammatory response [toll-like receptor 1 (*tlr1*), myeloid differentiation primary response 88 (*myd88*), nuclear factor kappa-light-chain-enhancer of activated B cells (*nfkb*), interleukin-1β (*il1b*), tumor necrosis factor alpha (*tnfa*), and interleukin-10 (*il10*)]. Both 60S ribosomal and β-actin were used as housekeeping genes. Data obtained were analyzed using the iQ5 optical system software version 2.0 (including GeneEx Macro iQ5 Conversion and GeneEx Macro iQ5 files; Bio-Rad). Primer sequences are reported in Table 2.

### 2.10. Physico-Chemical Characterization of Fish Fillets

The color of the fish skin and fillets (*n* = 15 per experimental group) were measured on triplicate positions (cranial, medial, and caudal) of the dorsal and of the epaxial region, respectively, and expressed as lightness (*L**), redness index (*a**) and yellowness index (*b**) according to CIE [87] with the CHROMA METER CR-200 (Konica Minolta, Chiyoda, Japan). Muscle pH was measured also on triplicate positions both in the right and left fillets. A pH-meter SevenGo SG2™ equipped with an Inlab puncture electrode (Mettler-Toledo, Schwerzenbach, Switzerland) was utilized. Then, the fillets were skinned and homogenized to ascertain the Water Holding Capacity (WHC) [88]. In brief, 2 g from the homogenized fillets were taken and inserted in plexiglass cylinders provided with a filter net; then, they were centrifuged at 1500 rpm (210 g) for 5 min. As a result, WHC was measured as the difference between the initial gross weight of cylinders and their gross weight after centrifugation divided by the fillet water content [68] and expressed as a percentage.

The total lipids content was obtained following the Folch’s method [71], gravimetrically quantified to express the result as g of lipid/100 g of fillet, and then suspended into 5 mL of chloroform. A quantity corresponding to 2.5 mg of total lipids were overnight trans-esterified to methyl esters (FAME) in 1% (*v*/*v*) sulfuric acid in methanol to determine the FA profile composition in the lipid extract [89]. After the methyl-esterification, the samples were injected into a Varian GC gas chromatograph (Varian Inc., Palo Alto, CA, USA) provided with a flame ionization detector and a Supelco Omegawax™ 320 capillary column (30 m × 0.32 mm × 0.25 μm) (Supelco, Bellefonte, PA, USA). The oven temperature started at 120 °C for 0.5 min and increased up to 170 °C at the rate of 10 °C/min, staying at this temperature for 6 min. Then, it rose to 220 °C at the rate of 3 °C/min and kept stable for 12 min. The duration of the run was 40 min for each sample. The injector and the detector temperatures were set at 220 °C and 300 °C, respectively. Helium was the carrier gas (constant flow of 1.5 mL/min), and the samples were injected in split mode (1:20) with an injection volume of 10 μL. Chromatograms were recorded with the Galaxie Chromatography Data System 1.9.302.952 (Varian Inc., Palo Alto, CA, USA). The FA were identified by comparing the peak retention time of the samples with the retention times of the fatty acid methyl ester standards (C4–C24, 18919-AMP, Sigma-Aldrich^®^, St. Louis, MO, USA). Then, they were individually quantified through calibration curves using tricosanoic acid (C23:0) (Sigma-Aldrich, St. Louis, MO, USA) as the internal standard. Data were expressed as a percentage of the total FAME.

### 2.11. Statistical Analyses

Data from growth parameters and somatic index, histological analysis, FTIR and relative quantification of gene expression were analyzed by one-way ANOVA followed by Tukey’s multiple comparison post hoc test (Prism 8; GraphPad software). Significance was set at *p* < 0.05. The assumption of normality was checked using the Shapiro–Wilk test. Furthermore, a simple linear regression coefficient was used to evaluate the correlations among fatty acids quantified in the present study.

Data from marketable indexes and physico-chemical analysis of fillets were analyzed by a one-way ANOVA followed by a post hoc Tukey test with the SAS statistical software [90]. A *p*-value of 0.05 was set as the minimum level of significance.

## 3. Results

### 3.1. Dietary Fatty Acid Composition and Antioxidant Component

The FA, tocopherols, and carotenoids profile are reported in Table 3. Considering the dietary FA content, the HM3 and HM20 diets highlighted a higher SFA content compared to the HM0 one, while the monounsaturated fatty acid (MUFA) content was characterized by a dose-dependent decrease from HM0 to HM20. PUFA and n-3 PUFA content was higher in HM20 with respect to the other diets. To conclude, both HM dietary inclusions reported a linear increment of n-3 PUFA (y = 0.4242x + 15.231; R² = 0.9974) and a decrement of MUFA (y = −0.615x + 38.909; R² = 0.7248) if compared to control. Finally, n-6 PUFA content was lower in the HM3 diet compared to the other ones.

The dietary HM inclusion led to a dose-dependent increase in lauric acid (C12:0) content. In all the experimental diets, palmitic acid (C16:0) was the most represented SFA. Considering MUFA, oleic acid (C18:1n-9) was higher in both HM diets, while a decreasing trend from HM0 to HM20 was highlighted for palmitoleic (C16:1n-9) and myristoleic (14:1) acids. Regarding PUFA, linoleic acid (LA, C18:2n-6) showed lower values in the HM3 and HM20 diets compared to the HM0 one. Eicosapentaenoic acid (EPA, C20:5n-3) content increased from the HM0 to the HM20 diet. Lastly, the HM20 diet showed a higher docosahexaenoic acid (DHA, C22:6n-3) content with respect to HM0 and HM3.

HM dietary inclusion also affected the tocopherol profile of the experimental diets. The HM0 diet did not contain α-tocopherol (Vitamin E equivalent), while both diets including HM highlighted an increment of α-tocopherol with the increase of the HM inclusion level, showing a linear trend (y = 0.8231x + 0.2759, R² = 0.9988) from 0 to 20%. The total amount of tocopherols highlighted a significant increasing trend from HM0 (14.43 ± 1.05 mg/kg) to HM20 (51.57 ± 1.98 mg/kg).

The results from the carotenoid analysis showed that only the HM20 diet contained both β-carotene and zeaxanthin (8.90 ± 1.98 and 3.38 ± 0.42 mg/kg for β-carotene and zeaxanthin, respectively).

### 3.2. Growth Performance, Condition Factor (K) and Marketable Characteristics

The fish growth performance parameters are reported in Table 4. At the end of the feeding trial, the survival rate of the fish was 100% in all the experimental groups. Final body weight (FBW), relative growth rate (RGR), specific growth rate (SGR), Fulton’s condition factor (K), and FCR were not impaired by dietary treatments (*p* > 0.05).

The results concerning the biometric and marketable characteristics are also presented in Table 4. Liver weight (LW), hepatosomatic index (HIS), carcass weight (CW) and viscerosomatic index (VSI) did not show significant difference among the experimental groups. However, the percentage of the fillet yield (FY) was significantly higher in the HM0 group compared to the others (*p* < 0.05).

### 3.3. Histological Analysis

No morphological or histopathological alterations were evident in the distal intestine samples from each experimental group (Figure 1). With regards to the histological indexes analyzed (Table 5), the HM20 group showed a significantly lower (*p* < 0.05) mucosal fold height compared to the HM0 group. Differently, no significant differences in submucosa width, supranuclear vacuoles and goblet cell abundance were observed among the experimental groups.

The results obtained by the liver histological analyses showed a physiological structure of the hepatic parenchyma (Figure 2).

### 3.4. FTIRI Analysis

Figure 3 shows the topographical distribution of lipids and proteins in the experimental groups. The false color images were calculated by using an arbitrary color scale. No relevant differences were observed in the relative amount of proteins and lipids among the experimental groups. Interestingly, a different distribution of these macromolecules was displayed in each group, suggesting a different cellular morphology within the tissue.

These findings were also confirmed by the statistical analysis of the LIP/TBM (relative amounts of lipids), FA/TBM (relative amounts of fatty acids) and PRT/TBM (relative amounts of proteins) band/area ratios (Figure 4). Lipids, fatty acids and proteins did not show significant differences among the experimental groups.

### 3.5. Gene Expression

Liver. The expression of *igf1*, *mstn1a* and *ghr1* (growth markers; Figure 5a–c), as well as *gr* and *hsp70* (stress response; Figure 5d,e) did not show significant differences among the experimental groups.

Pyloric caeca. Considering *elovl2* and *fads2* gene expression (lipid metabolism; Figure 6a,b), no significant differences were observed among the experimental groups.

Distal intestine. Concerning *il1b* gene expression, a significant (*p* < 0.05) upregulation was observed in the HM20 group compared to HM3 (Figure 7d). The results of the *il10* and *tlr1* gene expression (Figure 7a,f) evidenced significantly higher values (*p* < 0.05) in the HM20 group compared to HM0 and HM3, which did not show significant differences between them. Regarding *myd88* gene expression, the HM20 group showed a significant (*p* < 0.05) upregulation compared to the HM0 and HM3 ones (Figure 7b). Finally, considering the *nfkb* and *tnfa* gene expression (Figure 7c,e), no significant differences were detected among the experimental groups; however, a higher gene expression was detected in the HM20 group in both cases.

### 3.6. Physico-Chemical Traits of Fish Fillets

As shown in Table 6, the results related to the physical attributes of the fillets presented significant differences among the groups—except for the WHC, color parameters of skin and the lightness (L *) of fillet. Hence, the pH measurement of the fillet was significantly higher in HM3 (*p* < 0.05) than in the HM0 and HM20 groups, which showed similar values. In addition, both for the redness (a *) and the yellowness (b *) indexes, the highest scores were noted for the fillets of the HM3 treatment (*p* < 0.05).

The total lipid content (g/100 g of fillet) was affected by the different dietary regimes (*p* < 0.05), and the HM20 treatment presented the lowest lipid level (*p* < 0.05) (Table 7). In general, the FA profile of the fillets, which is also shown in Table 7, differed among the experimental groups. The SFA were significantly (*p* < 0.05) higher in the HM0 group, even if the lauric acid (C12:0) slightly increased in the HM3 and HM20 groups (*p* < 0.05). The sum of the MUFAs significantly (*p* < 0.05) increased following the dietary inclusion of the enriched HM. Moreover, the amount of n-6 PUFA significantly (*p* < 0.05) increased in the HM20 group, followed by HM3 and then HM0. Likewise, the n-3 PUFA levels rose in the HM20 group (*p* < 0.05) and then followed the previous trend. More specifically, the fillet FA composition revealed that the HM20 group was richer in linoleic acid (LA) compared to the other groups, which instead presented similar values. Notably, linolenic acid (ALA) was observed to be the major FA derived from the HM added to the HM20 formulation, while it was less present in the HM3 than in the HM0 one (*p* < 0.05). The quantity of eicosapentaenoic acid (EPA) did not significantly vary among the groups, whereas both docosapentaenoic acid (DPA) and docosahexaenoic acid (DHA) contents were higher in the HM20 and HM3 groups, respectively, compared to the HM0 one.

## 4. Discussion

The physiological responses of rainbow trout to different feed ingredients, including insect meal, have been deeply investigated over the last years [2,13,91,92,93]. However, no specific studies have been performed on rainbow trout fed diets containing insect meal obtained from black soldier fly prepupae cultured on a growth substrate enriched with spirulina. In the present study, an increasing content of total PUFA, n-3 PUFA, DHA and EPA from HM0 to HM20 diets was detected. The enrichment procedure of the insects’ biomass thus resulted in a progressive increase in important long-chain PUFA for fish dietary requirements in HM3 and HM20 diets, respectively, highlighting the suitability of this method. In fish, one of the major sites for the synthesis of long-chain PUFA, particularly DHA, is represented by pyloric caeca [85,94] that are also involved in the absorption of various lipid components (i.e., free lysophospholipids and monoacylglycerols) [95]. Two key classes of enzymes are involved in the synthesis of long-chain PUFA in fish: the FA elongase (Elovl) that catalyzes the extension of the FA chain of two carbons and the FA desaturase (Fads) that add a double bond to PUFA substrates [96]. Usually, elongase and desaturase genes are upregulated in fish when diets with low amounts of long-chain PUFA are provided in order to sustain the biochemical conversion of their precursors [97], as found by Bruni et al. [85]. In the present study, the expression of the genes related to fish-lipid metabolism analyzed in pyloric caeca did not show significant differences among the experimental groups, further confirming that all the diets were well balanced in terms of the PUFA profile.

In addition, the FA analyses on fish fillet revealed a comparable amount of n-3 PUFA among the experimental groups and a decreasing SFA content from the HM0 to HM20 groups. In this regard, the enrichment procedure used in the present study, together with the low dietary inclusions of HM, have possibly solved the well-known SFA-related drawback of insect-based diets; this is because, usually, the use of full-fat HM can determine a SFA increase in fish fillet with a parallel decrease of PUFA [98,99]. In fact, the addition of spirulina to the insect growth substrate increased the HM PUFA content, which was secondly transferred to the diets during the production activity. This led to comparable fillet LA, ALA, EPA and DHA content among the experimental groups.

As a confirmation of the proper formulation of the diets and their good acceptance by the fish, both the HM dietary inclusion levels did not affect growth performances (as also supported by the gene expression of growth-related markers) and marketable characteristics. These outcomes were expected since most of the literature does not indicate a growth reduction in rainbow trout fed diets including full-fat HM ranging from 10.5 to 20% [13,100].

When new ingredients are tested in aquaculture, it is essential to evaluate the gut health and its functional status, especially in terms of the micro-anatomical structure and expression of immune response-related markers. In this regard, histological analyses did not reveal major signs of inflammation and histopathological alterations, despite a significant reduction of the mucosal fold height observed in fish fed an HM20 diet compared to fish fed an HM0 one. Accordingly, a reduction in the absorptive epithelial surface (decrease of intestinal fold height) was observed in rainbow trout fed diets containing 25 or 50% full-fat HM with respect to the control FM-based diet by Cardinaletti et al. [13]. However, it should be pointed out that, despite the significantly lower value, the mucosal folds height showed by fish fed the HM20 diet was comparable to those observed in healthy rainbow trout analyzed in other studies [2,101].

Despite the good results obtained from the histological analyses, the expression of most of the immune response markers analyzed in the present study showed an upregulation in the HM20 group compared to the HM3 and HM0 ones.

This result may be possibly explained by analyzing the role of chitin. Lee at al. [102] stated that the action of chitin on the mammalian immune system is related to the polymer size. Larger chitin fragments (>70 μm) are inert; conversely, intermediate fragments (70–40 μm) can be recognized by TLR-2 possibly stimulating the TLR-2/MyD-88 signaling pathway with consequent cytokine production [103], while smaller fragments (<40 μm) can stimulate anti-inflammatory cytokine-like interleukin 10 production [102]. Additionally, Henry et al. [104], by analyzing the effects of diets including insect meal on *Dicentrarchus labrax*, suggested that the same polyhedric effect of different fragments of chitin is present in fish. TLR-2 works in synergy with TLR-1 [105], generating a complex that can be involved in the recognition of chitin fragments [106]. Although chitin fragment size was not measured in the present study, the natural digestion/degradation of chitin through the enzymatic activity could have generated fragments of a certain size responsible for the upregulation of certain immune-related markers observed in the fish fed the HM20 diet.

However, despite the upregulation of inflammatory markers, gut health was preserved in all the experimental groups, and this positive *scenario* can be attributed to specific molecules detected in HM diets. In this regard, the HM20 diet presented the highest amount of anti-inflammatory and antioxidant molecules, such as lauric acid (C12:0), tocopherols and carotenoids, with respect to the HM0 and HM3 ones. Specifically, several studies have shown that lauric acid, typically abundant in HM fat fraction, is a powerful anti-inflammatory molecule that can positively influence gut health [2,32]. However, the results of the present study highlighted a low amount of lauric acid in the HM20 diet (2.5%) if compared to other similar studies [107]; thus, the absence of macroscopic intestinal inflammation signs could be mainly related to other molecules such as tocopherols and carotenoids (β-carotene and zeaxanthin) which have been demonstrated to be successfully transferred from spirulina to insect and, secondly, to the feed in the present study.

It is widely known that animal organisms endogenously produce reactive oxygen species (ROS) during inflammatory processes [108], and the failure of the antioxidant systems to counteract their activities can result in prolonged inflammation, eventually developing the signs of a pathological condition [109]. Other authors have highlighted that carotenoids operate synergistically with tocopherols, acting as potent radical scavengers, thus preventing oxidative stress and inflammation [110,111,112]. In support to our hypothesis, these previous findings strongly suggest that the absence of inflammation observed in HM20 could be related to the action of these dietary antioxidant molecules.

However, it should be noted that, in addition to their role in cellular metabolism, carotenoids can also be deposited directly within fish chromatophore cells, thus coloring the skin and other tissues [110]. Carotenoids, together with tocopherols, have been demonstrated to be efficient antioxidants to preserve fish fillet quality, counteracting oxidation during cooking [75]. In the present study, most of the dietary antioxidant molecules were used by the HM20 fish to avoid the possible occurrence of inflammation in response to alternative ingredients, rather than to be deposited into the flesh as demonstrated by fillet color analysis. In fact, the highest scores for the redness (a*) and the yellowness (b*) indexes were observed in the HM3 group and not in the HM20 group—which, however, was fed the diet richest in these molecules. These data confirm the previous hypothesis that antioxidant molecules possibly exploit their antioxidant activity in preserving gut health and not acting as pigments in fish muscle when the gut is subjected to inflammatory events.

Finally, the liver is another organ that should be considered when new aquafeed ingredients are tested [113]. Several studies have demonstrated that the use of full-fat HM, at high dietary inclusion levels, can cause a severe steatosis in fish—mainly attributed to a high n-6/n-3 dietary *ratio* [114,115] and a consequent upregulation of genes involved in stress response [34]. Fish stress response is extremely important in aquaculture [116], and in recent years malnutrition has been listed as a possible stressor [117,118,119,120]. In the present study, no differences in the structure of liver parenchyma were observed among the experimental groups, and FT-IR analyses showed an absence of diversity in terms of total lipids, fatty acids and proteins. These results could be explained by the proper dietary formulation as well as by the n-6/n-3 dietary ratio, which was <1 in all the experimental diets. Consequently, *gr* and *hsp70* gene expression, analyzed in the same tissue, did not show differences among the experimental groups—confirming, in accordance with previous studies [13,33], the general state of health of the fish.

## 5. Conclusions

The present study demonstrated that lipids and bioactive molecules from spirulina, in particular PUFA, tocopherols and carotenoids, were successfully transferred to the fish diets by enriching the growth substrate of insects—demonstrating the suitability of the enrichment procedure proposed. In addition, this same method was essential for the use of a full-fat insect meal, avoiding the cost-effective defatting processes and preserving the important spirulina-derived molecules.

The results showed that some of these molecules possibly exploited their antioxidant activity in preserving the gut health in fish fed a HM20 diet, instead of acting as pigment in the fillet. Overall, growth performances, marketable characteristics and fish health were not negatively affected by the HM dietary inclusions, confirming the suitability of the diets proposed in the present study for rainbow trout culture.

## Figures and Tables

**Figure 1 animals-13-00173-f001:**
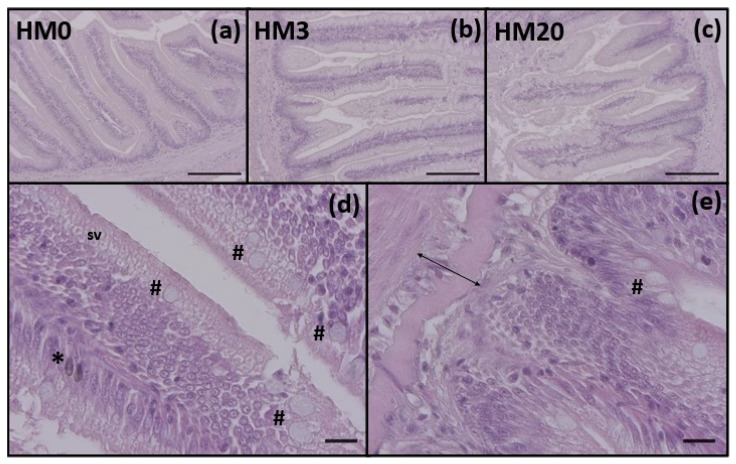
(**a**–**c**) Example of distal intestine histomorphology of rainbow trout fed experimental diets: (**a**) HM0, (**b**) HM3, and (**c**) HM20. (**d**,**e**) Details of mucosal folds (asterisk indicates melanomacrophages, hashtag indicate goblet cells, sv indicates supranuclear vacuoles and arrow indicate submucosa width). Scale bars: (**a**–**c**) = 200 μm; (**d**,**e**) = 20 μm.

**Figure 2 animals-13-00173-f002:**
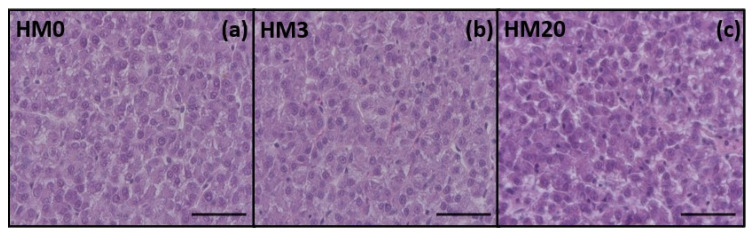
Representative histological images of liver parenchyma from rainbow trout fed experimental diets: (**a**) HM0, (**b**) HM3, and (**c**) HM20. Scale bars = 50 µm.

**Figure 3 animals-13-00173-f003:**
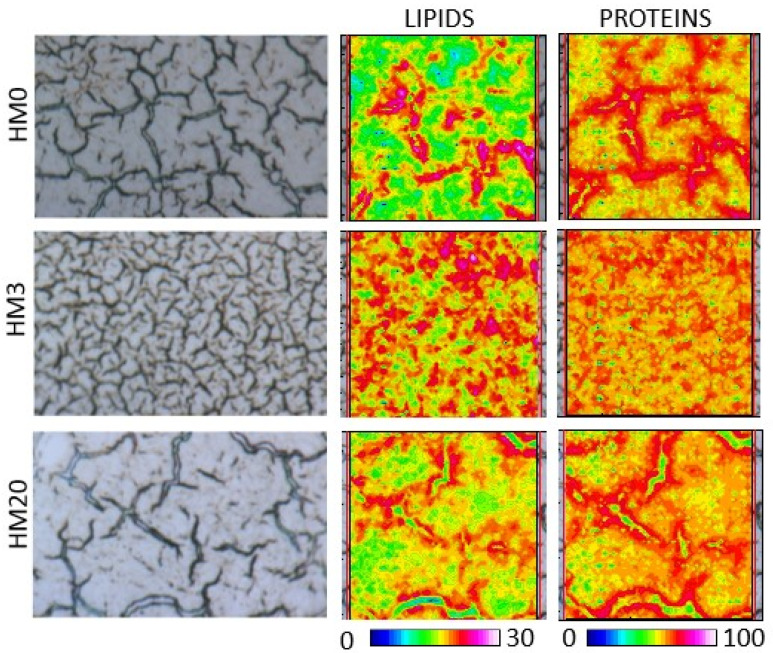
Hyperspectral imaging analysis of representative liver sample of rainbow trout fed the experimental diets. False color images show the topographical distribution of lipids (LIPIDS, 0–30 color scale) and proteins (PROTEINS, 0–100 color scale). Black/dark blue color represents the lowest absorbance values of the infrared radiation, while white/light pink the highest ones.

**Figure 4 animals-13-00173-f004:**
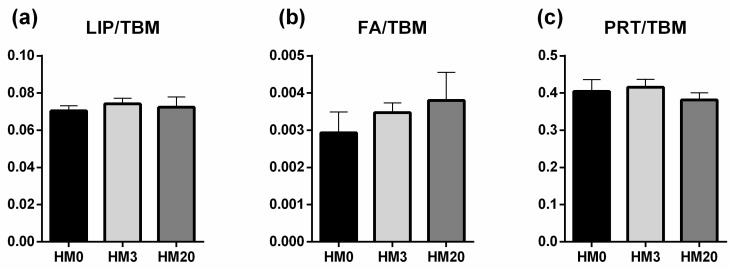
Biochemical composition of liver samples of rainbow trout fed the experimental diets. Statistical analysis of (**a**) LIP/TBM (relative amount of lipids), (**b**) FA/TBM (relative amount of fatty acids), and (**c**) PRT/TBM (relative amount of proteins) band area ratios was performed. Values are shown as mean ± SD (*n* = 9).

**Figure 5 animals-13-00173-f005:**
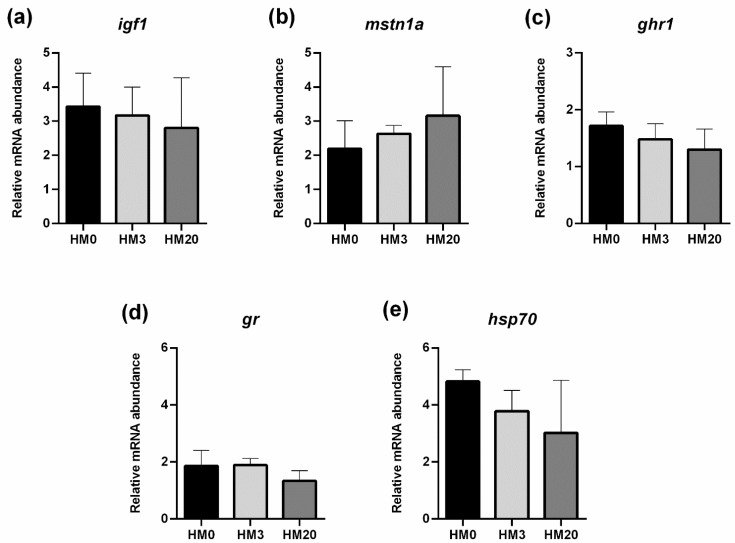
Relative mRNA abundance of genes involved in growth—(**a**) *igf1*; (**b**) *mstn1a*; (**c**) *ghr1*—and stress response—(**d**) *gr* and (**e**) *hsp70*—analyzed in liver samples of rainbow trout fed the experimental diets. Values are shown as mean ± SD (*n* = 9).

**Figure 6 animals-13-00173-f006:**
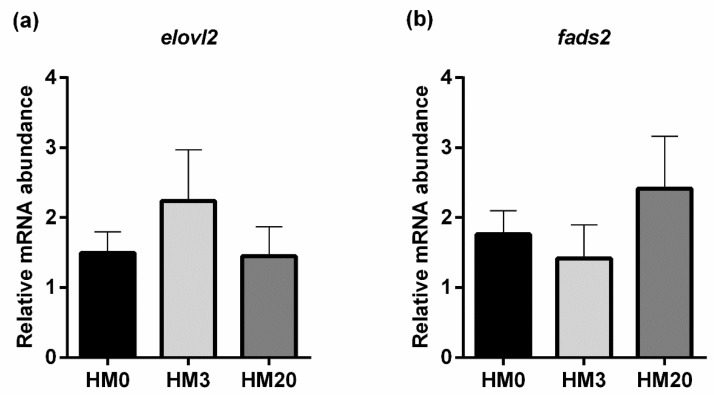
Relative mRNA abundance of genes involved in lipid metabolism, analyzed in pyloric caeca samples of rainbow trout fed the experimental diets. (**a**) *elovl2*, (**b**) *fads2*. Values are shown as mean ± SD (*n* = 9).

**Figure 7 animals-13-00173-f007:**
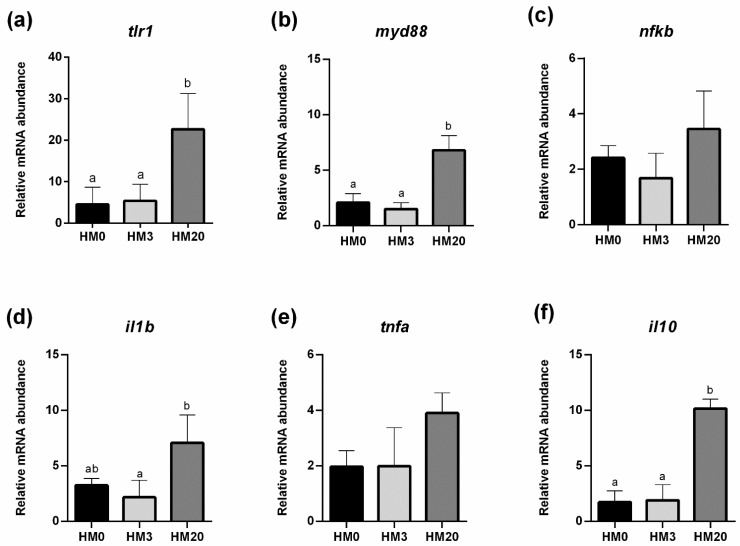
Relative mRNA abundance of genes involved in immune response, analyzed in distal intestine samples of rainbow trout fed the experimental diets. (**a**) *tlr1*, (**b**) *myd88*, (**c**) *nfkb*, (**d**) *il1b*, (**e**) *tnfa*, (**f**) *il10*. Values are shown as mean ± SD (*n* = 9). Different letters indicate significant differences among experimental groups (*p <* 0.05).

**Table 1 animals-13-00173-t001:** Ingredients (g/kg) and proximate composition (% DM) of HM and experimental diets.

	HM	HM0	HM3	HM20
Ingredients				
FM ^a^		420	404	307
Pea protein concentrate ^b^		120	120	120
HM		-	18	120
Wheat gluten meal ^a^		120	120	120
Wheat flour ^a^		128	128	130
FO ^a^		180	178	173
Soy lecithin ^a^		8	8	8
Mineral ^$^ & Vitamin ^#^ supplement		14	14	14
DL-Methionine		-	-	3
Binder ^c^		10	10	5
Proximate composition *				
DM	90.2	93.4	93.4	93.6
Total N	8.1	8.1	8.2	8.1
EE	21.0	23.2	23.8	23.6
Ash	15.2	8.8	8.9	9.0
Chitin ^+^	8.6		0.2	1.0
Gross Energy (MJ/kg)	23.2	24.9	25.1	25.0

^a^ Kindly provided by Skretting Italia, Mozzecane (VR, Italy); ^b^ Lombarda trading srl, Casalbuttano & Uniti (CR, Italy); ^c^ Sodium alginate (Merck KGaA, Darmstadt, Germany). ^$^ Mineral supplement composition (% mix): CaHPO_4_·2H_2_O, 78.9; MgO, 2.725; KCl, 0.005; NaCl, 17.65; FeCO_3_, 0.335; ZnSO_4_·H_2_O, 0.197; MnSO_4_·H_2_O, 0.094; CuSO_4_·5H_2_O, 0.027; Na_2_SeO_3_, 0.067. ^#^ Vitamin supplement composition (% mix): thiamine HCL Vit B1, 0.16; riboflavin Vit B2, 0.39; pyridoxine HCL Vit B6, 0.21; cyanocobalamin B12, 0.21; niacin Vit PP, 2.12; calcium pantothenate, 0.63; folic acid, 0.10; biotin Vit H, 1.05; myoinositol, 3.15; stay C Roche, 4.51; tocopherol Vit E, 3.15; menadione Vit K3, 0.24; Vit A (2500 UI/kg diet) 0.026; Vit D3 (2400 UI/kg diet) 0.05; choline chloride, 83.99. * Values are reported as mean of duplicate analyses. ^+^ Dietary chitin content was estimated starting from the chitin content of the test ingredient.

**Table 2 animals-13-00173-t002:** Primers and identification numbers. (HK) = housekeeping genes.

Genes	Forward Primer (5′-3′)	Reverse Primer (5′-3′)	ID Number	References
*ghr-1*	TGGAAGACATCGTGGAACCA	ATCAAAGTGGCTCCCGGTTA	AF403539	[86]
*igf1*	TGGACACGCTGCAGTATGTGTGT	CACTCGTCCACAATACCACGGT	GQ924783	[13]
*mstn1a*	CCGCCTTCACATATGCCAA	CAGAACCTGCGTCAGATGCA	AY839106	[13]
*gr*	AGAAGCCTGTTTTTGGCCTGTA	AGATGAGCTCGACATCCCTGAT	AY549305	[13]
*hsp70*	CCCTGGGCATCGAAACC	CCCTCGTAGACCTGGATCATG	AY423555	[13]
*elovl2*	TGGATGGGTCCCAGAGATGA	AGAAGGACAAGATCGTGAGGC	XM_036959634.1	[85]
*fads2*	GCCCTACCATCACCAACACC	AAACTCATCGACCACGCCAG	XM_036968520.1	[85]
*tlr1*	TGTTTGTCCTCTCTCGCCAC	CCCGTCTGTGTGGATAGACC	NM_001166101.1	[2]
*myd88*	GTTCCTGACGGTGTGTGACT	GTCGTTGGTTAGTCGTGTCC	NM_001124421.1	[2]
*nfkb*	AGCAACCAAACATCCCACCA	CTTGTCGTGCCTGCTTTCAC	XM_021614113.1	[2]
*il-1b*	ACATTGCCAACCTCATCATCG	TTGAGCAGGTCCTTGTCCTTG	NM_001124347.2	[2]
*tnf-α*	GGGGACAAACTGTGGACTGA	GAAGTTCTTGCCCTGCTCTG	AJ278085.1	[2]
*il-10*	CGACTTTAAATCTCCCATCGAC	GCATTGGACGATCTCTTTCTTC	NM_001245099.1	[2]
*60s* (HK)	TTCCTGTCACGACATACAAAGG	GTAAGCAGAAATTGCACCATCA	XM_021601278.1	[2]
*β-actin* (HK)	AGACCACCTTCAACTCCATCAT	AGAGGTGATCTCCTTCTGCATC	AJ438158.1	[2]

**Table 3 animals-13-00173-t003:** FA composition of the experimental diets expressed as percentage of total FA. Tocopherols and carotenoids are expressed as mg/kg of freeze-dried diet. Results are expressed as mean value ± standard deviation (*n* = 2).

	HM0	HM3	HM20
C12:0	-	0.40 ± 0.03	2.50 ± 0.10
C14:0	4.59 ±0.10	6.82 ± 0.21	6.75 ± 0.17
C14:1	6.15 ± 0.07	2.18 ± 0.02	-
C15:0	0.58 ± 0.03	0.81 ± 0.04	0.76 ± 0.01
C16:0	16.66 ± 0.27	26.39 ± 0.04	23.96 ± 0.31
C16:1n-9	22.10 ± 0.10	13.18 ±0.40	8.04 ± 0.07
C17:0	6.76 ± 0.01	2.37 ± 0.07	0.59 ± 0.02
C18:0	3.38 ± 0.04	5.06 ± 0.16	4.64 ± 0.06
C18:1n-9	11.21 ± 0.56	14.72 ± 0.14	15.76 ± 0.03
C18:1n-11	3.17 ± 0.28	2.61 ± 0.35	3.47 ± 0.96
C18:2n-6 (LA)	8.76 ± 0.06	7.84 ± 0.06	8.36 ± 0.07
C18:3n-6 (GLA)	1.20 ± 0.01	1.37 ± 0.08	1.42 ± 0.04
C18:3n-3 (ALA)	1.38 ± 0.14	1.48 ± 0.05	2.12 ± 0.27
C20:5n-3 (EPA)	6.25 ± 0.16	7.40 ± 0.18	11.02 ± 0.07
C22:5n-3 (DPA)	0.75 ± 0.06	0.63 ± 0.11	1.67 ± 0.05
C22:6n-3 (DHA)	7.07 ± 0.21	6.74 ± 0.48	8.94 ± 0.15
∑ SFA	31.97 ± 0.45	41.85 ± 0.15	39.21 ± 0.5
∑ MUFA	42.63 ± 0.66	32.69 ± 0.59	27.27 ± 0.86
∑ PUFA	25.4 ± 0.22	25.46 ± 0.74	33.53 ± 0.36
∑ n-3 PUFA	15.44 ± 0.15	16.25 ± 0.71	23.75 ± 0.25
∑ n-6 PUFA	9.96 ± 0.07	9.20 ± 0.03	9.77 ± 0.11
α-tocopherol	<LOD	3.07 ± 0.06	16.69 ± 0.82
γ-tocopherol	6.72 ± 0.44	7.77 ± 0.06	20.02 ± 0.31
δ-tocopherol	7.71 ± 0.62	6.55 ± 0.18	14.86 ± 0.85
Total tocopherols	14.43 ± 1.05	17.37 ± 0.18	51.57 ± 1.98
β-carotene	<LOD	<LOD	8.98 ± 1.98
zeaxanthin	<LOD	<LOD	3.38 ± 0.42

**Table 4 animals-13-00173-t004:** Growth performance parameters, condition factor (K), and marketable indexes of rainbow trout fed experimental diets.

	HM0	HM3	HM20
FBW (g)	42.92 ± 5.23	38.62 ± 5.85	40.91 ± 8.81
RGR (%)	189.8 ± 49.5	168.2 ± 49.1	175.2 ± 59.4
SGR (%/day)	2.5 ± 0.4	2.3 ± 0.4	2.3 ± 0.5
K	1.57 ± 0.11	1.57 ± 0.14	1.51 ± 0.16
FCR (%)	0.68 ± 0.05	0.70 ± 0.06	0.70 ± 0.05
FY (%)	54.26 ± 2.08 ^b^	52.46 ± 1.45 ^a^	52.21 ± 2.06 ^a^
LW (g)	0.505 ± 0.13	0.533 ± 0.10	0.530 ± 0.10
HSI (%)	1.180 ± 0.27	1.390 ± 0.23	1.317 ± 0.21
CW (%)	35.44 ± 1.44	34.56 ± 1.17	34.65 ± 1.64
VSI (%)	9.22 ± 1.06	9.33 ± 1.14	9.53 ± 1.24

Abbreviations: RGR, relative growth rate; SGR, specific growth rate; K, condition factor; FBW, final body weight; FY, fillet yield; LW, liver weight; HSI, hepatosomatic index; CW, carcass weight; VSI, viscerosomatic index. Carcass includes head, fins, skeleton, and skin. Different letters within a row indicate significant differences (*p* < 0.05). Data are reported as mean ± SD (*n* = 60 for growth performance parameters and K; *n* = 15 for marketable indexes).

**Table 5 animals-13-00173-t005:** Histological indexes (mucosal fold height, submucosa width, supranuclear vacuoles and goblet cell abundance) measured in distal intestine of rainbow trout fed experimental diets.

	HM0	HM3	HM20
Mucosal folds height (µm)	1165.4 ± 30.3 ^a^	1112.9 ± 74.2 ^ab^	1041.4 ± 81.3 ^b^
Submucosa width (µm)	33.88 ± 1.66	32.55 ± 0.83	33.81 ± 2.61
Goblet cells	9.2 ± 2.0	6.7 ± 3.2	7.8 ± 1.4
Supranuclear vacuoles	++	++	++

Values of mucosal folds length, submucosa width and goblet cells are shown as mean ± SD (*n* = 15). Scores: supranuclear vacuoles + = scattered, ++ = abundant. Different letters indicate significant differences among the experimental groups (*p* < 0.05).

**Table 6 animals-13-00173-t006:** Physical traits of the fillets from rainbow trout fed the three experimental diets.

	HM0	HM3	HM20
Skin color			
L*	52.63 ± 6.28	51.22 ± 5.72	52.15 ± 4.11
a *	−0.716 ± 1.03	−0.874 ± 0.95	−1.033 ± 0.47
b *	0.981 ± 1.83	0.687 ± 1.20	0.275 ± 1.21
Fillet color			
L*	49.04 ± 2.13	47.07 ± 1.98	50.01 ± 1.70
a *	2.55 ± 1.00 ^b^	4.02 ± 1.33 ^c^	0.87 ± 0.74 ^a^
b *	4.23 ± 0.79 ^a^	5.70 ± 0.50 ^b^	4.59 ± 0.88 ^a^
pH	6.49 ± 0.20 ^a^	6.73 ± 0.22 ^b^	6.50 ± 0.30 ^a^
WHC	96.81 ± 1.18	96.87 ± 0.74	96.73 ± 1.01

Data are reported as mean ± SD (*n* = 15). Different letters within the same row mean statistical difference (*p* < 0.05).

**Table 7 animals-13-00173-t007:** Total lipid (g/100 g) and FA composition of fillets from rainbow trout fed the three experimental diets, expressed as % of total FA. Results are presented as mean ± SD (*n* = 15). Different letters within the same row mean statistical difference (*p* < 0.05).

	HM0	HM3	HM20
Total lipid	3.985 ± 0.58 ^b^	3.880 ± 0.79 ^b^	3.352 ± 0.40 ^a^
Fatty acids			
C12:0	0.12 ± 0.01 ^a^	0.17 ± 0.01 ^a^	0.74 ± 0.18 ^b^
C14:0	3.47 ± 0.19 ^b^	3.38 ± 0.22 ^b^	2.73 ± 0.24 ^a^
C16:0	15.75 ± 0.49 ^c^	14.53 ± 0.52 ^b^	11.54 ± 0.76 ^a^
C16:1n-7	5.35 ± 0.28 ^a^	5.71 ± 0.28 ^b^	5.85 ± 0.45 ^b^
C18:0	3.78 ± 0.13 ^c^	3.46 ± 0.09 ^b^	2.94 ± 0.16 ^a^
C18:1n-9	18.07 ± 1.02 ^a^	18.13 ± 0.73 ^a^	19.72 ± 1.07 ^b^
C18:1n-7	2.88 ± 0.07 ^a^	2.96 ± 0.09 ^b^	2.92 ± 0.09 ^ab^
C18:2n-6 (LA)	9.45 ± 0.64 ^a^	9.49 ± 0.42 ^a^	10.69 ± 0.52 ^b^
C18:3n-3 (ALA)	1.74 ± 0.13 ^a^	1.71 ± 0.09 ^a^	1.88 ± 0.12 ^b^
C18:4n-3	1.39 ± 0.10 ^a^	1.46 ± 0.07 ^a^	1.57 ± 0.12 ^b^
C20:1n-9	1.55 ± 0.07 ^a^	1.54 ± 0.11 ^a^	1.70 ± 0.11 ^b^
C20:4n-6	1.08 ± 0.05	1.10 ± 0.06	1.07 ± 0.06
C20:5n-3 (EPA)	6.12 ± 0.39	6.38 ± 0.33	6.33 ± 0.60
C22:1n-11	1.06 ± 0.07 ^b^	1.07 ± 0.10 ^a^	1.27 ± 0.14 ^b^
C22:5n-3 (DPA)	1.50 ± 0.09 ^ab^	1.20 ± 0.72 ^a^	1.71 ± 0.11 ^b^
C22:6n-3 (DHA)	19.24 ± 1.60 ^a^	20.80 ± 2.09 ^b^	20.16 ± 1.33 ^ab^
Σ SFA	24.48 ± 0.74 ^c^	22.82 ± 0.79 ^b^	19.07 ± 1.05 ^a^
Σ MUFA	29.93 ± 1.19 ^a^	30.47 ± 1.17 ^a^	32.59 ± 0.90 ^b^
Σ n-6 PUFA	12.15 ± 0.69 ^a^	12.18 ± 0.45 ^a^	13.60 ± 0.58 ^b^
Σ n-3 PUFA	31.04 ± 1.70 ^a^	32.18 ± 1.85 ^ab^	32.61 ± 1.59 ^b^

SFA: saturated fatty acids; MUFA: monounsaturated fatty acids; PUFA: polyunsaturated fatty acids. The following FA, found below 0.5% of the total FAME, were utilized for calculating the Σ classes of FA but they are not listed: C13:0, C14:1n-5, isoC15:0, anteisoC15:0, C15:0, isoC16:0, C16:1n-9, C16:2n-4, C17:0, C16:3n-4, C17:1, C16:4n-1, C18:2n-4, C18:3n-6, C18:3n-4, C18:4n-1, C20:0, C20:1n-11, C20:1n-7, C20:2n-6, C20:3n-6, C20:3n-3, C20:4n-3, C22:0, C22:1n9, C22:1n-7, C22:2n-6, C21:5n-3, C22:4n-6, C22:5n-6, C24:0.

## Data Availability

Not available.

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
