# Peer review of "Spirulina-enriched Substrate to Rear Black Soldier Fly (Hermetia illucens) Prepupae as Alternative Aquafeed Ingredient for Rainbow Trout (Oncorhynchus mykiss) Diets: Possible Effects on Zootechnical Performances, Gut and Liver Health Status, and Fillet Quality"

_animals, 2023, doi:10.3390/ani13010173_

Round 1

Reviewer 1 Report

First of all, a lot of work was performed by a large research team of very qualified scientists, thus I feel obligated to perform a review which allows us to use the wide data set created in it. However, as a reviewer, I am obligated to give specific comments, which improve its soundness and quality. The title suggests that spirulina was added to Hermetia illucens meal – this is confusing in the case of described experimental design. Please reformulate the title to let know the reader that you mean Hermetia illucens meal prepared from larvae fed with dietary spirulina enrichment. In simple summary, you show elements of experimental design which are in opposition to the term described in the title as described above. At this point when I know that you used spirulina as a rearing substrate, I start to think – what is the benefit of enriching an insect diet vs. direct use of spirulina in a fish diet? Do you have any reason why would it be more efficient than direct use? It is hard to understand why to feed Hermetia which results in around 40% protein using spirulina which is characterized by 55-70% protein in its composition. Other important bioactive substances probably are not converted in a 1:1 ratio thus more efficient from a logical point would be to directly use spirulina in a fish diet For me, a key advantage would be if you show the composition of enriched and non-enriched Hermetia. Why 3 or 20%? I understand low and high and low inclusion ideas but why not 5 and 25%? Do you have any justification for those doses? Except for that using ratio of replacement due to various amounts of FM used among publications seems to be misleading. I would advise you to use shares in the diet as a better description thus 1,8 and 12% or 18 and 120 g/kg How did you assess welfare? Why there is no information about feed utilization? Line 96 – we can see the aim of the study, but there is no research hypothesis stated that should be presented for verification. The ethics statement is declared and no issues are recorded in this case. A good description of HI rearing conditions is given, however, the time of rearing is surprisingly high in comparison to the number of rearing days in large-scale production. Why did you lower the amount of binder in HM 20 diet? My experience says that the inclusion of HI meal especially full-fat which you used lowers the quality of feeds. Why did you use the nitrogen-to-protein ratio from the literature instead of calculating it on your own? The amino acid composition of HI meal and diets should be stated in the manuscript Are you sure that the proximate composition of the diets is based on analysis? If so crude protein should increase among diets with HI application due to the low KP ratio used in the experiment. I do not see the methodology of protein analysis. In terms of diet preparation, the extrusion method should be used – in another case (grinder) pellet quality will be low, and raw starch will lower the nutritive value of the experimental diet. You should perform a pellet quality assessment and analysis of the environmental sustainability and economic parameters of fish biomass production to give a full picture to the reader  - in the second case, it is crucial to keep performing the experiments as justified. Please look into: https://www.sciencedirect.com/science/article/abs/pii/S0044848620319311 https://pubmed.ncbi.nlm.nih.gov/33066664/ https://www.mdpi.com/2076-2615/10/11/2119 https://www.mdpi.com/2076-2615/11/3/604 https://sciendo.com/article/10.2478/aoas-2022-0071

https://sciendo.com/article/10.2478/aoas-2022-0070

Why you do not give any information on feed utilization, manuscript without FCR or FCE should be rejected and can not be assessed as fully describing the experiment. Please present this data and use it for further economic and sustainability calculations! The further part of the manuscript includes a brilliant set of data but it can not be assessed until all the above-mentioned issues will be described or fixed.

Author Response

Reviewer 1

First of all, a lot of work was performed by a large research team of very qualified scientists, thus I feel obligated to perform a review which allows us to use the wide data set created in it. However, as a reviewer, I am obligated to give specific comments, which improve its soundness and quality. 

The title suggests that spirulina was added to Hermetia illucens meal – this is confusing in the case of described experimental design. Please reformulate the title to let know the reader that you mean Hermetia illucens meal prepared from larvae fed with dietary spirulina enrichment.

Thank you very much for your suggestion, we decided to change the title as follows to make it clearer:

“Spirulina-enriched substrate to rear Hermetia illucens prepupae as alternative aquafeed ingredient for rainbow trout (Oncorhynchus mykiss) diets: possible effects on zootechnical performances, gut and liver health status, and fillet quality”.

In simple summary, you show elements of experimental design which are in opposition to the term described in the title as described above.

The title and the simple summary are now in accordance in terms of the elements of the experimental design.

At this point when I know that you used spirulina as a rearing substrate, I start to think – what is the benefit of enriching an insect diet vs. direct use of spirulina in a fish diet? Do you have any reason why would it be more efficient than direct use? It is hard to understand why to feed Hermetia which results in around 40% protein using spirulina which is characterized by 55-70% protein in its composition. Other important bioactive substances probably are not converted in a 1:1 ratio thus more efficient from a logical point would be to directly use spirulina in a fish diet 

The use of spirulina in the HI rearing substrate can be justified by different reasons, including economical ones. First of all, the suitability of this cyanobacterium as aquafeeds ingredient must be considered. In fact, the direct dietary inclusion of spirulina as fish meal replacer, especially at high levels, led to controversial results in several fish species (https://doi.org/10.1111/raq.12439). Thus, as suggested a possible solution could be a direct spirulina inclusion at lower concentrations, also in light to its high cost. As regards this last aspect, a recently published paper (http://dx.doi.org/10.1016/j.aquaculture.2022.738137) demonstrated that a 9% spirulina dietary inclusion was not able to counteract the negative side effects of completely vegetable diets intended for rainbow trout. At the same time other papers demonstrated that 1-15% dietary inclusion had positive effects in fish. However, the aim of the present study was to exploit the ability of HI larvae to modulate their fatty acid profile as well as their antioxidant molecules content according to the composition of the rearing substrate. This aim was selected in the view of using a full-fat insect meal avoiding the cost-effective defatting process which is regularly used for insect meal production.

For me, a key advantage would be if you show the composition of enriched and non-enriched Hermetia.

The major aim of the study was not to evaluate the modulation of the nutritional value of Hermetia illucens larvae by feeding spirulina-enriched coffee silverskin. For that reason, growth performance and conversion efficiency of the Hermetia illucens larvae are not reported here. We prefer to not report the original data of non-enriched and enriched Hermetia with spirulina, as the data will be deeply discussed in another specific research.

Why 3 or 20%? I understand low and high and low inclusion ideas but why not 5 and 25%? Do you have any justification for those doses? Except for that using ratio of replacement due to various amounts of FM used among publications seems to be misleading. I would advise you to use shares in the diet as a better description thus 1,8 and 12% or 18 and 120 g/kg

Regarding the substitution levels, the present study was aimed to test both the nutraceutical properties of HM (using 3% of HM respect to FM) and an ecological inclusion level (a global 20% FM substitution in aquafeeds has a proper ecological impact). This aspect has been clarified in the introduction section (lines 113-116 of the revised MS).

Considering your advice, we thought that was easier to name the experimental diets as HM3 and HM20 (so reporting the substitution level) instead of using the levels of inclusion. However, it we are agree that it is better specify also this information (that has been added in the methods section). The sentence, lines 142-146 of the revised MS, has been modified as follows: “Three complete diets were prepared to cover all the nutrient requirements for rainbow trout [66]: (i) a control diet (HM0) based on conventional marine (FM and FO) and vegetable ingredients; (ii) two test diets obtained from the HM0 formulation by replacing 3 (HM3) or 20 (HM20) % of FM and FO with HM (1.8 and 12 % levels of dietary inclusion, respectively)”

How did you assess welfare? 

“Welfare” could have not been the proper term to define the aim of the analyses performed in the present study. According to the previous suggestion to be consistent with the terms, we reformulate the sentence (lines 116-118 of the revised MS) as follows: “A multidisciplinary approach was the applied to evaluate fish zootechnical performances, gut and liver health status and marketable characteristics”.

Why there is no information about feed utilization? 

The reviewer is right suggesting the inclusion of feed utilization data. For that reason, we added the FCR formula in the methods section and the FCR results in Table 4.

In addition, regarding feed utilization, the sentence in the methods section (lines 225-229 of the revised MS) has been modified as follows: “For 6 weeks, fish fed the experimental diets provided in one daily meal in the morning until apparent satiety. All the feed provided was completely ingested by the fish within 30 minutes after feeding.”

Line 96 – we can see the aim of the study, but there is no research hypothesis stated that should be presented for verification.

The reviewer is right since there is a missing step in the sentence “These results, in addition to its high production costs [60], suggest that Arthrospira platensis, instead of an aquafeed ingredient or supplement in aquafeed formulation, could be used to enrich the Hermetia illucens growth substrate giving the possibility to exploit its beneficial properties avoiding a direct inclusion in fish diets”. This is due to the fact that the choice to use spirulina derived from a previous study (unfortunately not published yet and thus not cited) in which some of the authors of the present MS demonstrated that the inclusion of 15% of spirulina in the HI rearing substrate represented the best compromise between the amount of spirulina (economical aspect) and the best improvements in terms of prepupae nutritional profile. In support to this statement, we can provide the title of the oral presentation in which these results were shown:

Nartea, A. et al, 2021. “Modulation of nutritional value of Hermetia illucens larvae by feeding microalgae-enriched coffee silverskin: amino acids, fatty acids, and carotenoids” oral presentation at 18th Euro Fed Lipid Congress and Expo, online, October 18-21.

The ethics statement is declared and no issues are recorded in this case. A good description of HI rearing conditions is given, however, the time of rearing is surprisingly high in comparison to the number of rearing days in large-scale production.

The reviewer is right to state that the time of rearing is higher compared to that observed in large-scale production. However, the prepupae used in the present study were reared within our laboratories, without the large-scale production equipment, and this inevitably led to a longer rearing time (which is in line with previous studies).

Why did you lower the amount of binder in HM 20 diet? My experience says that the inclusion of HI meal especially full-fat which you used lowers the quality of feeds.

We agree with Reviewer comment about the fact that the inclusion of full-fat HM could result in less stable pellets. However, we understand that the process for producing the diet needs to be implemented and for this, in the revised version of the MS this part has been corrected as follows:

“All ingredients were ground (0.5 mm) with a Retsch Centrifugal Grinding Mill ZM 1000 (Retsch GmbH, Haan, Germany) and well mixed with FO to form a homogeneous blend (Kenwood kMix KMX53 stand Mixer). The resulting meshes were added with water (~350 g/kg) and the doughs thus obtained were cold extruded into 3 mm pellets using a meat mincer provided with a knife at the die. The wet pellets were then dried in a ventilated oven at 37 °C for 48 h”.

This procedure resulted in very stable pellets in water. We did some preliminary test to establish whether a reduction in the inclusion level of the binder could affect pellet stability, but we did not find any difference by comparing diets H20 vs H3 and H0 in terms of pellet swelling time and complete dissolution in static water.

Why did you use the nitrogen-to-protein ratio from the literature instead of calculating it on your own? 

Yes, the reviewer rises a good point. Unfortunately, at the time of submission the amino acid composition of HM and diets were not available.

This subject is always disputed in papers when the CP value of the HM are published or going to be published.  In case of full fat HM different Kp values have been reported by various authors (e.g. 4.67 by Janssen et al. 2017 and 5.12 by Rawski et al., 2020). Because of this, in the new version of the MS we preferred to report the total N content of HM and diets instead of using questionable CP value.

The amino acid composition of HI meal and diets should be stated in the manuscript 

From literature, it is well known that insect meal for HM is a good protein source and has a good amino acid profile. For this reason, the amino acidic composition was not evaluated. However, the diets were formulated to cover all known nutrient requirements of trout (NRC 2011) by considering theoretical values for the amino acid composition of all ingredients. For this reason, at the highest level of insect meal inclusion, the HM20 diet was supplemented with 3g/kg of DL-methionine in order to cover the amino acid needs of the trout.

Are you sure that the proximate composition of the diets is based on analysis? If so crude protein should increase among diets with HI application due to the low KP ratio used in the experiment.

Yes, we are sure. The N content of the test diets was analyzed with the Kjeldahl method reference (AOAC 2005).

It is not clear why the dietary protein content should increase among diets with graded HM inclusion. The opposite would be the case since HM (8.07% total N) was included in spite of FM, the latter ingredient being higher in total N (11%) than HM. This substitution is expected to result in a slight decline in N (CP) content rather than an increase.

I do not see the methodology of protein analysis.

Please see previous answer. Crude protein contents of HM and test diet were calculated from the analyzed total N contents as detailed in previous answers.

In terms of diet preparation, the extrusion method should be used – in another case (grinder) pellet quality will be low, and raw starch will lower the nutritive value of the experimental diet. You should perform a pellet quality assessment and analysis of the environmental sustainability and economic parameters of fish biomass production to give a full picture to the reader  - in the second case, it is crucial to keep performing the experiments as justified. Please look into:

https://www.sciencedirect.com/science/article/abs/pii/S0044848620319311

https://pubmed.ncbi.nlm.nih.gov/33066664/

https://www.mdpi.com/2076-2615/10/11/2119

https://www.mdpi.com/2076-2615/11/3/604

https://sciendo.com/article/10.2478/aoas-2022-0071

https://sciendo.com/article/10.2478/aoas-2022-0070

Yes, the reviewer is right in the advisable use the extrusion process to produce aquafeeds, but unfortunately, we did not do it in the present study. Moreover, as previously stated, about diet preparation, the revised version of the MS has been implemented. In addition, we also agree in the fact that the raw starch source used could have affected (lowered) the nutritional value of the test diets. However, based on the quite high growth rate attained by fish with all diets after 6 weeks feeding, it seems that fish well tolerated the offered diets.

As previously mentioned, a preliminary test of the pellets was conducted to evaluate stability (flaking / dissolving) in static water, without observing changes among diets which could eventually affect the results of the feeding trial.

We agree with the reviewer that an analysis of the environmental sustainability and economic parameters of fish biomass production will definitely provide a full picture to the reader, but it would require long lasing experiment where fish should attain the commercial size. While in this experiment, diet response was assessed in fish at the juvenile fish stage, and it was mainly focused on fish physiological responses. We are sorry for this and will consider this suggestion for further studies

Why you do not give any information on feed utilization, manuscript without FCR or FCE should be rejected and can not be assessed as fully describing the experiment. Please present this data and use it for further economic and sustainability calculations! The further part of the manuscript includes a brilliant set of data but it can not be assessed until all the above-mentioned issues will be described or fixed.

These final remarks were added to the main text as explained above.

Reviewer 2 Report

The MS has a huge amount of information on growth performace and physical effects, named welfare in the title, of dietary inclusion of HI meal form prepupae enriched with spirulina. From my pont of view, the term welfare in the title is confusing and maybe other words can describe better the aims of the study: health, immune response, ...Also, an important part of the paper deal with the composition and physical traits of the fillets and nothing is said in the title.

My main concern about the MS is related to the tested diets. First, the authors stated that the inclusion of spirulina in fish diets is only feasible at low levels (1 to 15%) and also that it has high production costs. Thus, in this study is proposed an indirect use of spirulina to enrich H. illucens, being neccesary to steps: first to produce spirulina (it is included 15% in the substrate) and, afterthat , to rear H.illucens prepupae and obtain meal (HM). Nothing is dicussed about the economical benefit to do it instead to use spirulina as an ingredient in fish diets. Maybe a combination of spirulina and non-enriched HM could be more interesting?

Moreover HM3 diet have a low FM replacement (only 16 g kg) and only 18 g kg of HM were included. This led to the same question, why does not make a direct use of spirulina?. From my point of view a high level of HM should be more adequate to formulate the intermediate diet.

The HM20 diet was supplemented with DL-Methionine, without any explanation for this. This supplementaion could mask the results obtained.

In the discussion th authors stated that the content od DHA in fillets increased in HM20 diet (lines 524 to 527) but non significant differences were found compared to the control.

To conclude, the hypothesis and experimental design has some important gaps that need to be corrected.

Author Response

Reviewer 2

The MS has a huge amount of information on growth performace and physical effects, named welfare in the title, of dietary inclusion of HI meal form prepupae enriched with spirulina. From my pont of view, the term welfare in the title is confusing and maybe other words can describe better the aims of the study: health, immune response, ...Also, an important part of the paper deal with the composition and physical traits of the fillets and nothing is said in the title.

Thank you very much for your suggestion. According also to the comment of R1, we decided to change the title as follows: “Spirulina-enriched substrate to rear Hermetia illucens prepupae as alternative aquafeed ingredient for rainbow trout (Oncorhynchus mykiss) diets: possible effects on zootechnical performances, gut and liver health status, and fillet quality”.

My main concern about the MS is related to the tested diets. First, the authors stated that the inclusion of spirulina in fish diets is only feasible at low levels (1 to 15%) and also that it has high production costs. Thus, in this study is proposed an indirect use of spirulina to enrich H. illucens, being neccesary to steps: first to produce spirulina (it is included 15% in the substrate) and, afterthat , to rear H.illucens prepupae and obtain meal (HM). Nothing is dicussed about the economical benefit to do it instead to use spirulina as an ingredient in fish diets. Maybe a combination of spirulina and non-enriched HM could be more interesting? Moreover HM3 diet have a low FM replacement (only 16 g kg) and only 18 g kg of HM were included. This led to the same question, why does not make a direct use of spirulina? From my point of view a high level of HM should be more adequate to formulate the intermediate diet.

The use of spirulina in the HI rearing substrate can be justified by different reasons, including economical ones. The aim of the present study was to exploit the ability of HI larvae to modulate their nutritional profile according to the composition of the rearing substrate with emphasis on fatty acid composition and antioxidant molecules. HI has a good protein content and amino acid profile, but it is usually subjected to defatting procedures before aquafeed formulations (due to its unbalanced fatty acid profile) which represents an important manufactured cost. In this context, the use of microbial biomass to enrich the HI rearing substrate has proved to be a valid solution to improve the insect PUFA content (http://dx.doi.org/10.1016/j.anifeedsci.2019.114309; http://dx.doi.org/10.1016/j.aquaculture.2019.734659). Similarly, in the present study, the use of full-fat HI prepupae meal reared on an organic substrate enriched with spirulina: (i) allowed to avoid the insect defatting procedure, saving additional costs for prepupae meal preparation, due to the high PUFA content of this cyanobacterium; (ii) aimed to investigate also the possible transfer of spirulina-derived bioactive molecules that, if transferred to insect meal during the larval growth, can exert a beneficial role on fish health. The use of low amounts of spirulina to enrich the insect substrate, coupled with the good results in terms of fish growth, overall health and marketable features, represents a valid and innovative solution that can amortize the high cost of this raw material within the aquaculture sector.

As regards the use of spirulina (15 % W/W) it represents the best compromise between the amount of spirulina (economical aspect) and the best improvements in terms of prepupae nutritional profile. In each case, these results are reported in an oral presentation titled “Modulation of nutritional value of Hermetia illucens larvae by feeding microalgae-enriched coffee silverskin: amino acids, fatty acids, and carotenoids” presented online at 18th Euro Fed Lipid Congress and Expo, October 18-21

The HM20 diet was supplemented with DL-Methionine, without any explanation for this. This supplementaion could mask the results obtained.

Although it is now well known that insect meal for HM is a good protein source and has a good amino acid profile, it is equally true from the literature that this ingredient presents some amino-acid defects in particular as regards the methionine level when compared to fishmeal. The diets were formulated to cover all known nutrient requirements of trout (NRC 2011) by considering theoretical values for the amino acid composition of all ingredients. This led us to integrate, at the highest level of insect inclusion, the HM20 diet with 3g/kg of DL-methionine in order to cover the amino acid needs of the trout. In this case the authors are sure that the results obtained are valid.

In the discussion the authors stated that the content of DHA in fillets increased in HM20 diet (lines 524 to 527) but non significant differences were found compared to the control.

Thanks for reporting our error. The sentence (line 597-599 of the revised MS) has been corrected as follows: “In fact, the addition of spirulina to the insect growth substrate increased the HM PUFA content which was secondly transferred to the diets during the production activity. This led to comparable fillet LA, ALA, EPA, and DHA content among the experimental groups.”

Reviewer 3 Report

The research was focused on the effect of Spirulina fed black solider fly(Hermetia illucens) on the growth performance, fillet biochemistry, gut histology, lipid metabolism and immune system. The research was appropriately designed and the MS is suitable to be published on Animals

However, there are still few questions:

1. Data analysis. In the MS, the number of sample used in the analysis or the number of fish that used in the analysis was record as "n" in the analysis, for instance, n=2 in Table 3, n=15 in Table 6 and 7. However, the "n" in the One-Way ANOVA analysis should be independent samples. The duplicates in the diet analysis was not independent but from the same diet sample. Thus, One-way ANOVA could not be used in Table 3. Similarly, for the analysis of fish samples, each tank should be viewed as one unit in One-way ANOVA analysis instead of each fish in the tank (Nutrient Requirements of Fish and Shrimps, 2011).

2. In the introduction, the authors showed that low inclusion of spirulina in fish diet is able to improve fish growth, survival rate and feed intake, but when the dietary level is high, feed acceptance can be reduced (line 85-91).  In present study, the spirulina fed black solider fly did not show statistically improvement  in growth performance and gut histological analysis even when the replacement level of fishmeal was low (HM3 group). Thus, does it mean that it could be better if spirulina is directly used in the diet other than using spirulina fed black solider fly? If possible, could the authors add the amunot of spirulina used in the black solider fly feeding?

Author Response

Reviewer 3

The research was focused on the effect of Spirulina fed black solider fly(Hermetia illucens) on the growth performance, fillet biochemistry, gut histology, lipid metabolism and immune system. The research was appropriately designed and the MS is suitable to be published on Animals

However, there are still few questions:

  1. Data analysis. In the MS, the number of sample used in the analysis or the number of fish that used in the analysis was record as "n" in the analysis, for instance, n=2 in Table 3, n=15 in Table 6 and 7. However, the "n" in the One-Way ANOVA analysis should be independent samples. The duplicates in the diet analysis was not independent but from the same diet sample. Thus, One-way ANOVA could not be used in Table 3. Similarly, for the analysis of fish samples, each tank should be viewed as one unit in One-way ANOVA analysis instead of each fish in the tank (Nutrient Requirements of Fish and Shrimps, 2011).

Thank you for highlighting this technical mistake. Repeated measure of the same diet sample should be analyzed by a repeated-measure one-way ANOVA followed by the Tukey’s multiple comparison post hoc test. We specified this in the section regarding the statistical analyses. However, even performing this analysis, the significance among groups did not vary, so the Table 3 remains unaltered.

Differently, fish cannot be considered as repeated measures of the same tank due to the biological and physiological variability of each specimen (especially if we consider the gene expression). We know that fish in the same tank are subjected to same environmental conditions, but reducing the results obtained for each fish as a simple mean value would eliminate the above-mentioned variability that instead must be considered, especially in fish nutrition studies.

https://doi.org/10.1016/j.aquaculture.2021.737351

https://doi.org/10.1038/s41598-020-80379-x

  1. In the introduction, the authors showed that low inclusion of spirulina in fish diet is able to improve fish growth, survival rate and feed intake, but when the dietary level is high, feed acceptance can be reduced (line 85-91).  In present study, the spirulina fed black solider fly did not show statistically improvement  in growth performance and gut histological analysis even when the replacement level of fishmeal was low (HM3 group). Thus, does it mean that it could be better if spirulina is directly used in the diet other than using spirulina fed black solider fly? If possible, could the authors add the amunot of spirulina used in the black solider fly feeding?

The aim of the present study was not to confirm the possible use of spirulina as FM replacer, since starting from the available literature, it has been demonstrated that the direct inclusion (at low levels) of this cyanobacterium in fish diets led to several benefits. Differently, we would suggest another method to exploit the beneficial properties of low amounts of spirulina (compensating its high production costs) and this was the enrichment of the HI growth substrate. In this sense, spirulina is intended as a possible solution to improve the nutritional quality of this insect which is known to possess low amounts of long-chain PUFA. By improving the insect PUFA content no defatting procedures are necessary and manufacturing cost are significantly reduced. Additionally, important bioactive molecules that can exert beneficial role on fish health are also maintained in the full fat insect meal. As regards the use of spirulina (15 % W/W) it represents the best compromise between the amount of spirulina (economical aspect) and the best improvements in terms of prepupae nutritional profile. In each case these results are reported in an oral presentation titled “Modulation of nutritional value of Hermetia illucens larvae by feeding microalgae-enriched coffee silverskin: amino acids, fatty acids, and carotenoids” presented online at 18th Euro Fed Lipid Congress and Expo, October 18-21

Considering the amount of spirulina used for HI feeding, we cannot report the exact amount of spirulina used to feed the quantity of prepupae actually used to produce the dietary HM, since we produced more prepupae than we needed for fish diet preparation.

Round 2

Reviewer 1 Report

The manuscript has been revised and supplemented according to my suggestions.

Author Response

Reviewer 1

The manuscript has been revised and supplemented according to my suggestions

Many thanks.

Reviewer 2 Report

Answers provided by the authors are in general included in the corrected MS. However, still there are some points to be considered before the paper will be published.

The term welfare still appear in some parts of the MA (i.e. abstract). 

From my point of view, the explanation about levels of HI included: "The two selected inclusions levels were chosen because 20 % represents an ecological inclusion level (a global 20% FM substitution in aquafeeds has a proper ecological impact) while the 3% substitution level should be considered as a feed supplement inclusion with possible ameliorative effects on fish health status"  is too weak. Why a 20% of FM substitution has a proper ecological impact and not 25% or any other percentage?

Author Response

Reviewer 2

Answers provided by the authors are in general included in the corrected MS. However, still there are some points to be considered before the paper will be published.

The term welfare still appear in some parts of the MA (i.e. abstract).

The term welfare has been substituted along the MS.

From my point of view, the explanation about levels of HI included: "The two selected inclusions levels were chosen because 20 % represents an ecological inclusion level (a global 20% FM substitution in aquafeeds has a proper ecological impact) while the 3% substitution level should be considered as a feed supplement inclusion with possible ameliorative effects on fish health status"  is too weak. Why a 20% of FM substitution has a proper ecological impact and not 25% or any other percentage?

As reported in the review article now included in the revised MS (Ref. 62), extensive studies have been conducted on rainbow trout and Hermetia illucens meal over the last years. This recent review reports most of these papers, and the HM inclusion levels  ranges from 3 to 100%. Considering these results, a 20 % of substitution cannot represent a nutraceutical inclusion level (in these case levels are usually < or equal to 10%) and thus a more obvious consideration can be the ecological one. In accord to the reviewer also higher inclusion levels can of course have the same significance. However, as properly described by this recent review, several studies have been performed using different percentages of FM substitution with HM in rainbow trout, different from the canonical 25 or 50% (suggested by the reviewer), without using any justification about their choice, supporting the hypothesis that this research area still needs deeper knowledge and certain degree of freedom in dietary formulation. In each case, the 20% selected in the present work has been previously selected in other papers, amnd is thus in line with other studies that are summarized in the reported review.

Reviewer 3 Report

Thanks for the response from the authors. Repeated-measure One-Way ANOVA should be performed if 3 or more related groups are significantly different from each other. However, the diets used in the research were independent from each other, thus repeated-measure One-Way ANOVA cannot be used for diet analysis. In addition, the conference that the author quoted in the response did not support the answer. In the first reference (https://doi.org/10.1016/j.aquaculture.2021.737351), One-way ANOVA did not performed in the analysis of the diet, but in the analysis of fish samples. In the second reference, even thought they used 9 sturgeon for the analysis, the authors marked out n=3 in the Methods. Each fish in one unit can be viewed as one sample when the fish was tagged individually.

Author Response

Reviewer 3

Thanks for the response from the authors. Repeated-measure One-Way ANOVA should be performed if 3 or more related groups are significantly different from each other. However, the diets used in the research were independent from each other, thus repeated-measure One-Way ANOVA cannot be used for diet analysis. In addition, the conference that the author quoted in the response did not support the answer. In the first reference (https://doi.org/10.1016/j.aquaculture.2021.737351), One-way ANOVA did not performed in the analysis of the diet, but in the analysis of fish samples. In the second reference, even thought they used 9 sturgeon for the analysis, the authors marked out n=3 in the Methods. Each fish in one unit can be viewed as one sample when the fish was tagged individually.

We decided to remove the statistical analysis from the table 3, according to several studies (https://doi.org/10.1016/j.aquaculture.2019.734596; https://doi.org/10.1111/anu.12574; https://doi.org/10.1186/s40104-017-0191-3) that simply report the means of the dietary fatty acid profile and according to the reviewer that suggests to not use the ANOVA since the n have to be independent samples. Accordingly, both the table 3 and the result section were revised. Thank you for your suggestion.

Considering the n used for fish, we agree with the reviewer that in the first paper reported in the previous answer (https://doi.org/10.1016/j.aquaculture.2021.737351) the authors used the ANOVA for the analyses of fish samples and not for the fatty acid profile of the diets. However, this paper was previously reported only to highlight the fact that for analyses on fish samples, the authors used a n=10 (although they had 3 tanks per experimental group).

In the same way, in the paper related to the sturgeon reported in the previous answer (https://doi.org/10.1038/s41598-020-80379-x), the number of fish reported in the methods (9 or 15 depending on the analyses) was exactly the same number of fish used for the statistical analyses (as reported in the caption of tables and figures). The n=3 reported in the methods was an alternative way to indicate that the number of fish derived from 3 tanks (suggested by a reviewer but not agreed by us). Some authors use the tanks as experimental unit only for zootechnical data (growth and other biometric indexes), but for the analyses related to the fish physiological responses, individual specimens are considered as the experimental unit. The reason has been already explained in the previous answer, but we want to highlight this point in a better way. For certain analyses, fish should be considered as experimental unit due to the biological and physiological variability of each specimen (especially if we consider the gene expression). We know that fish in the same tank are subjected to same environmental conditions, but reducing the results obtained for each fish as a simple mean value would eliminate the above-mentioned variability that instead must be considered, especially in fish nutrition studies.

Here we report some studies, published on eminent journals, that adopted this method:

https://doi.org/10.1016/j.aquaculture.2019.734220

https://doi.org/10.1038/s41598-019-45172-5

https://doi.org/10.1016/j.aquaculture.2021.736550

https://doi.org/10.3390/ani12131698

https://doi.org/10.1016/j.aquaculture.2019.734539

https://doi.org/10.1016/j.aquaculture.2021.737132

https://doi.org/10.1016/j.aquaculture.2013.10.035

https://doi.org/10.1016/j.aquaculture.2007.09.028

For these reasons, in order to avoid the loss of significance of our results, and in accord to our statistical expert, we suggest accepting our option which is well documented by the literature.